# Effect of Sheath Blade Removal on *Phyllostachys violascens* Shoot Quality

**Sen Xu, Shuanglin Chen, Ziwu Guo \*, Yuyou He, Liting Yang, Yawen Dong, Yanyan Xie and Jingrun Zhang**

Research Institute of Subtropical Forestry, Chinese Academy of Forestry, Hangzhou 311400, China
\* Correspondence: guoziwu@caf.ac.cn; Tel.: +86-571-63139580

**Abstract:** Sheath blades are the first bamboo organ exposed to sunlight after shoots initially emerge. However, it remains uncertain whether sheath blades affect bamboo shoot growth and quality. Accordingly, this study explores variations in *Phyllostachys violascens* shoot growth and quality, comparing natural growth conditions to periodic sheath blade removal treatments. Results show that sheath blade removal and interactions between sheath blade removal and duration had no significant effect on the morphology, protein nutritional quality, or value of bamboo shoots. However, the length of bamboo shoot was significantly lower 4 d after treatment compared to 4 d after natural growth conditions. Moreover, sheath blade removal did have a significant effect on soluble sugar, total acid, oxalic acid, tannic acid, and cellulose content as well as sugar–acid ratios of bamboo shoots, while having no significant effect on the content and proportion of amino acid flavor compounds. Interactions between sheath blade removal and duration only had a significant effect on total acid and sugar–acid ratios. Soluble sugar, oxalic acid, tannic acid, and sugar–acid ratios increased significantly 2 d after sheath blade removal, while total acid and cellulose content decreased significantly. Lastly, soluble sugar content and sugar–acid ratios increased significantly 4 d after sheath blade removal. Findings from this study indicate that sheath blades affect shoot quality, particularly taste, which is mainly driven by carbon metabolism, but the effect of nitrogen metabolism was not obvious. This study gave a new perspective for revealing the formation mechanism of shoot quality, and also provided possible methods of improvement for the shoot quality.

**Keywords:** *Phyllostachys violascens* shoots; sheath blade removal; height growth; nutritional quality; taste quality

## 1. Introduction

Bamboo shoots are an important resource output of bamboo forests, while also being China's bulk export agricultural product. With the increasing demand for high-quality bamboo shoots in domestic and foreign markets, the nutritional value and taste quality of bamboo shoots have become important factors that restrict their economic value and market potential [1,2]. Improving or maintaining the quality of bamboo shoots is not only an important objective of bamboo forest management but also an important way to improve the economic benefits of bamboo forests. Developmental mechanisms associated with bamboo shoot quality has long been an important research topic. However, most relevant studies have focused on the impact of environmental factors and human management interference (i.e., soil nutrient management, soil mulching cultivation, and stand regulation) on bamboo shoot quality and associated developmental mechanisms [3–5]. In other words, the potential influence of bamboo shoot organs on quality has been largely ignored.

Generally, water and nutrient uptake by roots and leaf assimilation work interdependently to maintain plant growth and metabolism [6,7]. Rowland et al. [8] reported that leaf shape had an important effect on tomato quality and orbicular leaves had a significant positive effect on fructose content and tomato yields. Wang et al. [9] also reported that high chlorophyll content in flag leaves during the yellow ripening stage was beneficial to

the formation of rice taste quality, but it inhibited protein synthesis. It therefore stands to reason that plant leaves are closely associated with formative crop quality. However, bamboo shoots have no branches or leaves; they are only attached by a culm sheath. Moreover, sheath blades are situated at the top of the culm sheath, which is an important bamboo classification feature. Different bamboo species vary greatly in shape (triangle, cone, lanceolate, etc.), size, and color [10]. After the elongation, the bamboo internode is complete, and sheath blades generally dry up and fall off along with sheaths. As a unique organ of bamboo shoots, whether the physiological function of sheath blades is the same or similar to that of leaves has yet to be confirmed scientifically. Sheath blades are also the first bamboo organ exposed to sunlight after shoots initially emerge. Additionally, many studies have shown that shoot quality changes considerably prior to and following the emergence of sheath blades [11]. However, it remains uncertain whether sheath blades are an important organ affecting the variable quality of bamboo shoots.

*Phyllostachys violascens* is an economically important bamboo species in many provinces of South China. It is easy to plant and experiences rapid growth, high yields, and superior bamboo shoot quality [12]. Its sheath blades are lanceolated to zonal, inverting, and strongly corrugated. Our previous study showed that pigment compounds of sheath blades were closely related to bamboo shoot taste quality. Taste quality is positively correlated to chlorophyll and negatively correlated to carotenoids. Analysis showed that sheath blades may potentially be an important organ affecting shoot quality through light signal transduction pathways [13]. However, no relevant studies have systematically explained these phenomena or reported similar results. To further clarify the relationship between sheath blades and shoot quality, this study used *Ph. violascens* shoots as the study object, applying the cutting method to remove sheath blades after bamboo shoots were unearthed. This study analyzes differences in height growth, nutritional value, and taste quality between sheath blade removal treatments and natural (untreated) conditions during different periods. The objective of this study is to answer the following question: Is there an obvious positive or negative change in *Ph. violascens* shoot quality (morphology and nutrition, flavor, and rough substances content) during different periods following sheath blade removal? Answering this question will help us to better understand developmental mechanisms associated with bamboo shoot quality, while providing a theoretical reference for the cultivation of high-quality bamboo shoots.

## 2. Materials and Methods

### 2.1. Experimental Site

The experiment was conducted at the Niutou Mountain National Forest Park, Pingyao Town, a division of Yuhang District (119°40′–120°23′ E, 30°09′–30°34′ N), Hangzhou City, Zhejiang Province, China, which is situated at the end of the Tianmu Mountain system. This region is characterized by a subtropical monsoon climate with an annual precipitation of 1150–1550 mm. Its average annual temperature is 16.2 °C, while the annual average temperature in January is 3.8 °C and the annual average temperature in July is 28.5 °C. The extreme maximum high temperature is 40.5 °C, and the extreme minimum low temperature is −11.6 °C. The annual average sunshine hours = 1970 h with 244 frost-free days. Lateritic red soil is the soil type. The site and sampling plot is an important testing base of our laboratory. Thus, no license was required for us to conduct the experiment and sampling.

### 2.2. Experimental Method

A 0.5 hm$^2$ high yield *Ph. violascens* forested area was selected for our experiment (density: 17,040 $\pm$ 315 stem/hm$^2$; diameter at breast height (DBH): 5.04 $\pm$ 0.41 cm; age structure: 1a:2a:3a = 1.82:1.27:1). In total, 180 newly unearthed bamboo shoots were randomly selected within the experimental forest area and marked with conspicuously designed tags that were placed next to bamboo shoots. Sheath blades were removed from 90 bamboo specimens (Figure 1a), after which newly formed sheath blades were removed before 8:00 each day until sampling was complete (Figure 1b,c). The remaining

90 bamboo specimens were used as the control (i.e., untreated). Shoots were sampled on 26 and 28 March 2021, namely 2 and 4 d after sheath blades were removed. By day 6, after bamboo shoots were excavated, they were too mature (tall) to be consumed; consequently, sampling was terminated. For each treatment, 30 bamboo shoots were used, 10 being used as replicates where three separate repetitions were conducted. Fresh shoots were then placed in an icebox and brought back to the laboratory for analysis.

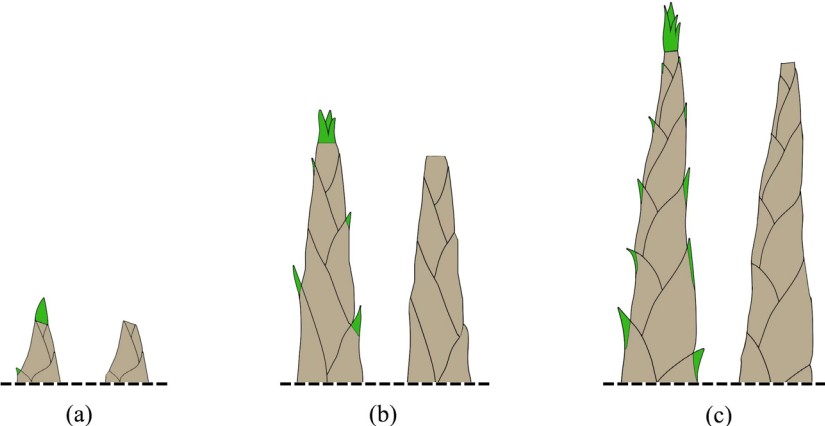

**Figure 1.** Schematic diagram of sheath blades removal: (**a**) treatment of sheath blade removal; (**b**) sheath blade removal after 2 d; and (**c**) sheath blade removal after 4 d. Pictures (**a–c**) show naturally growing bamboo shoots on the left and sheath blades removed bamboo shoots on the right.

*2.3. Measurements*

An electronic balance (SE202F, OHAUS Corporation, Parsippany, NJ, USA) was used to measure the individual specimen weight. Base diameter (mm) and length (cm) were measured using a vernier caliper and ruler, respectively. For each sample, the shoot shell was peeled, the part with a high degree of lignification at the base was removed and weighed using the aforementioned electronic balance, the weight was recorded as edible weight, and then the edible rate was calculated (edible rate = (edible weight/individual specimen weight) × 100). Next, a grinder was used to crush the bamboo shoots. A homogenate suspension was used to determine protein, starch, soluble sugar, total acid, oxalic acid, tannic acid, and free amino acid. A proportion of the fresh samples was heated at 60 °C to constant weight in an oven, after which it was ground into a powder and filtered through a 0.150 mm mesh to determine cellulose and lignin content.

*2.4. Determination of Physiological Indicators*

Protein content was determined by the Kjeldahl method. Added to the homogenate suspension (5.0 g) was 0.4 g copper sulfate, 6.0 g potassium sulfate, and 20 mL sulfuric acid for digestion. When the temperature of the digester reached 420 °C, the suspension was allowed to continue to digest for 1 h. When the liquid was green and transparent, 50 mL of water was added after cooling. The nitrogen content was measured by an automatic Kjeldahl nitrogen analyzer (Foss Kjeltec 2300, Hilleroed, Denmark), and the protein content of the sample was calculated.

Soluble sugar and starch content were determined by the anthrone "colorimetric" method. A total of 15 mL of 80% ethanol was added to 0.2 g of dried sample, which was then heated for 10 min in boiling water and centrifuged at $5000 \times g$ for 10 min. The supernatant was collected by extracting it repeatedly three times, then fixing the volume to 50 mL. We took 0.2 mL of extracted solution and added 5 mL of anthrone, then heated it at 90 °C for 15 min, measured the absorbance at 620 nm, and calculated the soluble sugar content. Subsequently, precipitate was added (10 mL of 30% perchloric acid) and left overnight. Then, we put it in 80 °C water for 10 min and centrifuged at $4000 \times g$ for 10 min. The supernatant was fixed to a volume of 50 mL, the absorbance was determined according to the aforementioned method, and the starch content was calculated.

Oxalic acid content was determined by ion-exclusion chromatography with refractive index and diode array detectors as per Mo et al. [14]. A total of 30 mL of 0.05% sulfuric acid solution was added to 5.0 g of homogenate suspension to homogenize for 2 min (RH Basic 2, Staufen, Germany), the homogenizer was cleaned with 10 mL of 0.05% sulfuric acid, the cleaning solution was combined, and the volume was fixed to 50 mL with 0.05% sulfuric acid. The sample after 5 mL of constant volume was centrifuged at $10,000 \times g$ at 4 °C for 10 min, and the supernatant was passed through a 0.22 μm water system needle filter to be tested. We used a Waters Alliance E2695 (Milford, MA, USA) liquid chromatograph with a chromatographic column Aminex HPX-87H, $7.8 \times 300$ mm, with a mobile phase of 0.1% sulfuric acid solution, column temperature at 35 °C, flow rate set to 0.60 mL/min, and an injection volume of 10 μL. Diode array detector (DAD) detector wavelengths were 210 nm and 254 nm. The differential refractive detector (RID) detector temperature was 35 °C.

Total acid content was determined by titration. Homogenate suspension (200.0 g) was dissolved in the same amount of boiling water as the sample. The solution was filtered with filter paper and the filtrate was collected. After the addition of 60 mL of water and 0.2 mL of 1% phenolphthalein indicator in 50 g of filtrate, we titrated the solution with 0.1 mol/L of sodium hydroxide until it was reddish for 30 s, then recorded the volume of the 0.1 mol/L of sodium hydroxide to calculate the total acid content.

Tannic content was determined by the titration "spectrophotometry" method. A total of 80 mL of water was added to the homogenate suspension (5.0 g), which was heated in boiling water bath for 30 min. The extracted solution was mixed with water to a total volume of 100 mL. Then, 2 mL of solution was centrifuged at $8000 \times g$ for 4 min. We absorbed 1 mL of supernatant, adding 5 mL of water, 1 mL of mixed solution of sodium tungstate and sodium molybdate, and 3 mL sodium carbonate solution (75 g/L); the absorbance was measured at 765 nm after 2 h, and then the tannin content was calculated.

Free amino acid content was measured using an amino acid analyzer. A total of 60 mL of water was added to the homogenate suspension (5.0 g), which was extracted in a boiling water bath for 1 h, then cooled down to room temperature. The extracted solution was filtered to a 100 mL capacity bottle, and the residue was washed twice with 20 mL of deionized water. The volume was fixed after the combination of the washing solution and the extract. We took 5 mL of solution from it, added 5 mL of 10% sulfosalicylic acid solution, then centrifuged at $10,000 \times g$ for 15 min in 4 °C. The supernatant was for 0.22 μm membrane to be tested. The samples were analyzed on an amino acid analyzer (YLSZJ-SB-175); the preparation of mobile phase was same as that of Mo et al. [15] and a gradient elution was applied. The injection volume was 20 μL and the flow rate of the buffer solution and ninhydrin was 0.35 mL/min. The calculation method of the flavor amino acid content was as follows: umami amino acid = Asp + Glu; bitter amino acid = Val + Ile + Leu + Tyr + Phe + Tyr; aromatic amino acid = Phe + Tyr; and sweet amino acid = Gly + Thr + Ala + Pro + Ser.

Cellulose and lignin content were determined by sulfuric acid hydrolysis [16]. A total of 70 mL of neutral detergent fiber (pH = 7.0) was added to 0.1 g of dried sample, which was held at 100 °C for 40 min and 115–121 °C for 20 min. It was washed with 95% ethanol and absolute ethanol two times through the filter to reach pH 6.5–7.0. The residue was placed in a vacuum desiccator and dried for 20 min. The residue was put into 70 mL of 2 mol/L HCL and held at 100 °C for 50 min. The residue was washed with 95% ethanol, absolute ethanol, and acetone twice with the filter until it was neutral. The residue was baked at 80 °C to a constant weight, denoted as $W_1$. A total of 10 mL of 72% sulfuric acid was added to the residue. After degradation at 20 °C for 4 h, 90 mL of distilled water was added and it was stored overnight at room temperature. Then, the residue was washed with distilled water to pH 6.5 and dried until constant weight, recorded as $W_2$. The residue was ashed in a muffle furnace at 550 °C and dried in a desiccator to a constant weight, recorded as $W_3$. Finally, the cellulose and lignin content were calculated: cellulose = $W_1 - W_2$; lignin = $W_2 - W_3$.

*2.5. Statistical Analysis*

SPSS Statistics (ver. 25.0) was selected to conduct Student's *t*-tests, which were used to analyze the effects of sheath blade removal on the morphology, nutrition, and taste quality of *Ph. violascens* shoots. All tested data are means ± SD (n = 3).

Essential amino acids (*EAA*s) of bamboo shoot proteins were analyzed according to the standard method recommended by FAO/WHO [17], and results were expressed as amino acid scores (*AAS*s):

$$AAS = \frac{A_x}{A_s} \times 100 \tag{1}$$

where $A_x$ is the *EAA* content of the bamboo shoot proteins being tested and $A_s$ is the recommended *EAA*s of FAO/WHO 2007. Recommended *EAA*s of FAO/WHO 2007 are as follows: Thr, 23; Val, 39; Ile, 30; Leu, 59; Lys, 45; Met + Cys, 22; Phe + Tyr, 38.

The essential amino acid index (*EAAI*) of bamboo shoot proteins was calculated as follows:

$$EAAI = \sqrt[n]{\frac{T_{Thr}}{S_{Thr}} \times \frac{T_{Val}}{S_{Val}} \cdots \times \frac{T_{Phs+Tyr}}{S_{Phs+Tyr}}} \times 100 \tag{2}$$

where *n* was the number of *EAA*s used for comparison (i.e., *n* = 7 in this study); T is the *EAA* content of bamboo shoot proteins being tested; and S is the *EAA* content of protein in eggs. The *EAA* content of protein in eggs is as follows: Thru, 40; Val, 50; Ile, 40; Leu, 70; Lys, 55; Met + Cys, 35; Phe + Tyr, 60.

The nutritional index (*NI*) of bamboo shoot proteins was calculated as follows:

$$NI = \frac{EAAI \times PP}{100} \tag{3}$$

where *PP* is the percentage of proteins in the sample being tested.

The closeness degree (*CD*) of bamboo shoot proteins was calculated as follows:

$$CD = 1 - C\sum_{K=1}^{7} \frac{|a_k - u_{ik}|}{|a_k + u_{ik}|} \times 100 \tag{4}$$

where $a_k$ is the *EAA* content of proteins in eggs (Table 1); $u_{ik}$ is the *EAA* content of the bamboo shoot proteins being tested; and *C* is a constant (*C* = 0.09 in this study).

**Table 1.** Statistical two-way ANOVA results of sheath blade removal on morphological and edible rates of *Ph. violascens* shoots.

| Indexes | Sheath Blade Removal | | Duration | | Sheath Blade Removal × Duration | |
|---|---|---|---|---|---|---|
| | *F* | *p* | *F* | *p* | *F* | *p* |
| Length | 3.871 | 0.085 | 178.912 | 0.000 | 2.846 | 0.130 |
| Base diameter | 2.170 | 0.179 | 2.880 | 0.128 | 0.043 | 0.842 |
| Individual weight | 3.857 | 0.085 | 29.879 | 0.001 | 0.309 | 0.593 |
| Edible rate | 0.551 | 0.479 | 1.087 | 0.328 | 0.003 | 0.960 |

## 3. Results

*3.1. Effects of Sheath Blade Removal on Bamboo Shoot Morphology and Edible Rates*

As shown in Table 1, duration significantly affected shoot length and individual weight, while sheath blade removal and interactions between sheath blade removal and duration had no significant effects on basic morphological and edible rates. After sheath blade removal, the length, base diameter, and individual weight of bamboo shoots all decreased. Moreover, there was an obvious decrease in length (5.39%) 4 d after treatment (Figure 2). As the treatment period was extended, the length and individual weight increased significantly,

while no significant changes were observed in the base diameter and edible rates. This implies that sheath blade removal inhibits the height growth rate of bamboo shoots.

**Figure 2.** Morphological and edible rates after *Ph. violascens* sheath blade removal. CK$_2$: natural growth after 2 d; BR$_2$: sheath blade removal after 2 d; CK$_4$: natural growth after 4 d; BR$_4$: sheath blade removal after 4 d; ns: $p > 0.05$; *: $p < 0.05$; **: $p < 0.01$; ***: $p < 0.001$.

*3.2. Effects of Sheath Blade Removal on Bamboo Shoot Nutritional Quality*

Duration significantly affected protein, total amino acid, and *EAA* content in bamboo shoots, but sheath blade removal and interactions between sheath blade removal and duration had no significant effect on nutritional quality (Table 2). During the same treatment period, protein, starch, total amino acid, and *EAA* content as well as the proportion of *EAA*s in bamboo shoots after sheath blade removal all increased or decreased by varying degrees, but differences were not significant (Figure 3). Total amino acid and *EAA* content decreased significantly 4 d after sheath blade removal and natural growth, while protein content decreased significantly 4 d after sheath blade removal. However, duration had no significant effect on starch and *EAA* ratios.

**Table 2.** Statistical two-way ANOVA results after *Ph. violascens* sheath blade removal on shoot nutritional quality.

| Indexes | Sheath Blade Removal | | Duration | | Sheath Blade Removal × Duration | |
|---|---|---|---|---|---|---|
| | *F* | *p* | *F* | *p* | *F* | *p* |
| Protein | 0.045 | 0.837 | 7.755 | 0.024 | 0.826 | 0.390 |
| Starch | 0.483 | 0.507 | 1.342 | 0.280 | 0.029 | 0.869 |
| Total amino acid | 2.266 | 0.171 | 70.631 | 0.000 | 0.343 | 0.574 |
| Essential amino acid | 1.645 | 0.236 | 59.220 | 0.000 | 0.885 | 0.374 |
| Proportion of essential amino acid | 0.340 | 0.576 | 0.085 | 0.778 | 0.766 | 0.407 |

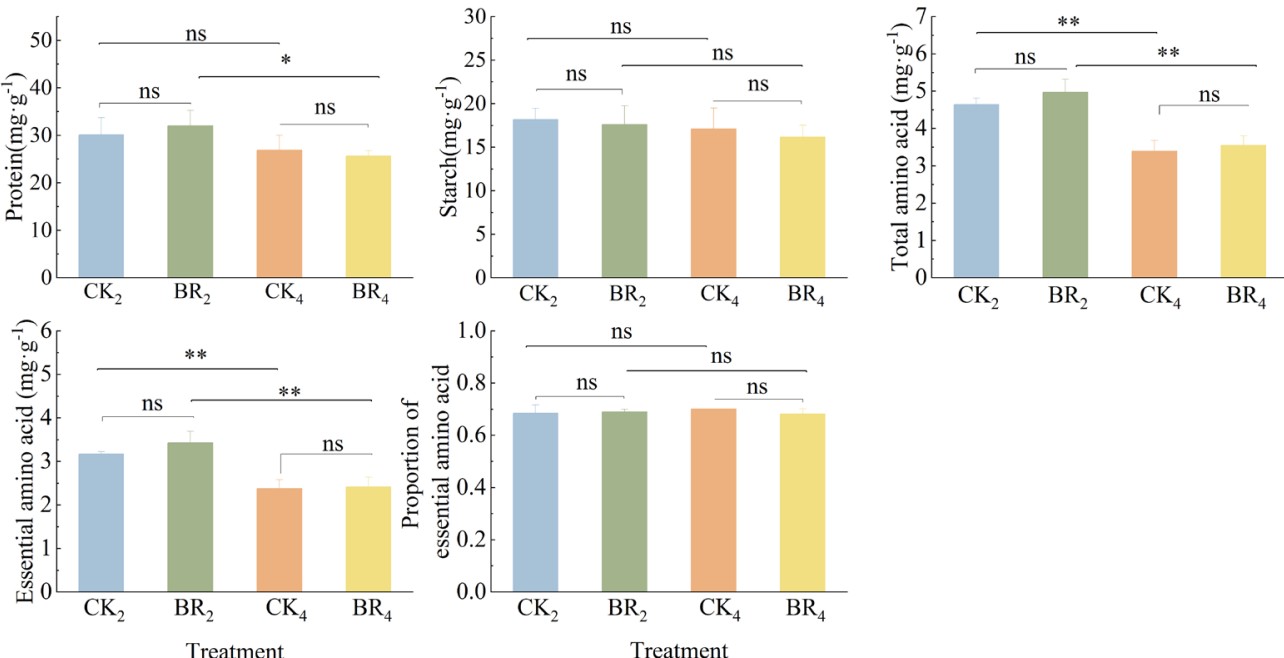

**Figure 3.** Nutritional quality of *Ph. violascens* shoots after sheath blade removal. CK$_2$: natural growth after 2 d; BR$_2$: sheath blade removal after 2 d; CK$_4$: natural growth after 4 d; BR$_4$: sheath blade removal after 4 d; ns: $p > 0.05$; *: $p < 0.05$; **: $p < 0.01$.

### 3.3. Effects of Sheath Blade Removal on Bamboo Shoot Protein Nutritional Values

Duration significantly affected valine, methionine + cystine, phenylalanine + tyrosine, total *EAA* scores, and *CD* of bamboo shoots, while sheath blade removal and interactions between sheath blade removal and duration had no significant effect on protein nutritional values (Table 3). After sheath blade removal, the total *EAA* score increased. Except for phenylalanine + tyrosine, the AAS of the other factors ranged from 12.41 to 30.80, which was significantly lower compared to the standard FAO/WHO *EAA* pattern (Figure 4a). Egg protein was used as the standard to further evaluate bamboo shoot protein nutritional values by calculating the *CD*, *EAAI*, and *NI*. The results show that *CD* and *NI* values increased after sheath blade removal (Figure 4b). Compared to the control (untreated), however, *EAAI* decreased after treatment day 2 and increased after treatment day 4. The total *EAA* score, *CD*, *EAAI*, and *NI* significantly decreased as duration increased.

**Table 3.** Statistical two-way ANOVA results of *Ph. violascens* sheath blade removal on shoot nutritional values.

| Indexes | Sheath Blade Removal | | Duration | | Sheath Blade Removal × Duration | |
|---|---|---|---|---|---|---|
| | *F* | *p* | *F* | *p* | *F* | *p* |
| Thr score | 0.287 | 0.607 | 3.324 | 0.106 | 0.069 | 0.799 |
| Val score | 0.719 | 0.421 | 35.673 | 0.000 | 1.531 | 0.251 |
| Ile score | 1.971 | 0.198 | 0.733 | 0.417 | 0.847 | 0.384 |
| Leu score | 1.801 | 0.216 | 0.032 | 0.862 | 0.104 | 0.755 |
| Lys score | 1.392 | 0.272 | 3.916 | 0.083 | 0.009 | 0.928 |
| Met + Cys score | 0.014 | 0.908 | 6.504 | 0.034 | 1.920 | 0.203 |
| Phe + Tyr score | 1.062 | 0.333 | 19.078 | 0.002 | 0.011 | 0.919 |
| Total essential amino acid score | 0.839 | 0.386 | 5.970 | 0.043 | 0.002 | 0.967 |
| Essential amino acid index | 0.815 | 0.393 | 0.554 | 0.478 | 0.137 | 0.721 |
| Nutrition index | 1.920 | 0.203 | 4.935 | 0.057 | 1.330 | 0.282 |
| Closeness degree | 0.901 | 0.370 | 6.090 | 0.039 | 0.000 | 1.000 |

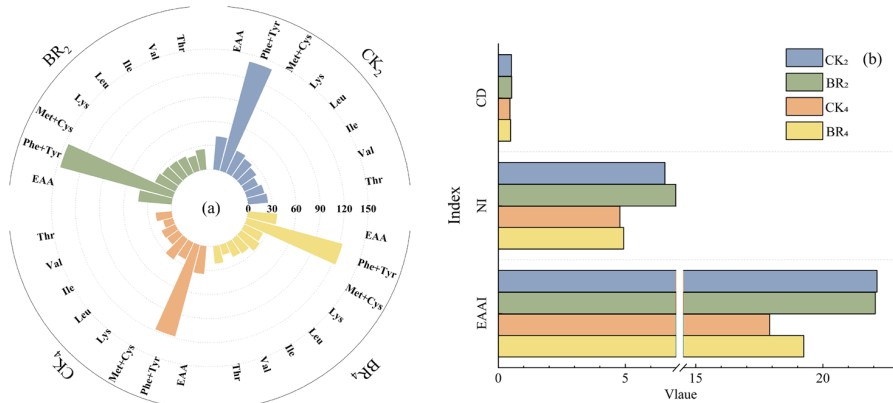

**Figure 4.** Nutritional quality of *Ph. violascens* shoots after sheath blade removal. (**a**): essential amino acids (*EAA*s) of bamboo shoot proteins; (**b**): *EAAI, NI, CD* of bamboo shoot proteins; $CK_2$: natural growth after 2 d; $BR_2$: sheath blade removal after 2 d; $CK_4$: natural growth after 4 d; $BR_4$: sheath blade removal after 4 d.

### 3.4. Effects of Sheath Blade Removal on the Taste Quality of Bamboo Shoots

Sheath blade removal significantly affected soluble sugar, total acid, oxalic acid, tannic, and cellulose content as well as the sugar–acid ratio of bamboo shoots, while duration significantly affected lignin, bitter, and sweet aromatic amino acid flavor compounds. However, the interactions between sheath blade removal and duration only significantly affected the total acid content and the sugar–acid ratio (Table 4). Soluble sugar, oxalic acid, and tannic content as well as the sugar–acid ratio of bamboo shoots increased significantly 2 d after sheath blade removal, while total acid and cellulose content decreased considerably (Figure 5). Soluble sugar content and the sugar-acid ratio of bamboo shoots increased significantly 4 d after sheath blade removal, but no significant differences were observed among the other indexes. Compared to 2 d after sheath blade removal, total acid, bitter, and aromatic and sweet amino acid compounds of bamboo shoots decreased significantly, while lignin content and the sugar–acid ratio increased significantly 4 d after the treatment. However, compared to 2 d of natural growth, bitter, aromatic, and sweet amino acids compounds significantly decreased, while lignin content significantly increased 4 d after natural growth. It can therefore be concluded that sheath blade removal significantly affects bamboo shoot taste quality, but there were differences among durations.

**Table 4.** Statistical two-way ANOVA results of sheath blade removal on flavor and roughness substances in *Ph. violascens* bamboo shoots.

| Indexes | Sheath Blade Removal | | Duration | | Sheath Blade Removal ×Duration | |
|---|---|---|---|---|---|---|
| | *F* | *p* | *F* | *p* | *F* | *p* |
| Soluble sugar | 14.357 | 0.005 | 1.210 | 0.303 | 2.525 | 0.151 |
| Total acid | 22.794 | 0.001 | 4.071 | 0.078 | 9.464 | 0.015 |
| Sugar–acid ratio | 29.887 | 0.001 | 1.715 | 0.227 | 7.289 | 0.027 |
| Oxalic acid | 9.961 | 0.013 | 0.330 | 0.581 | 1.735 | 0.224 |
| Tannin | 6.470 | 0.035 | 0.078 | 0.787 | 0.267 | 0.620 |
| Cellulose | 9.467 | 0.015 | 0.333 | 0.580 | 3.404 | 0.102 |
| Lignin | 0.855 | 0.382 | 19.324 | 0.002 | 0.001 | 0.979 |
| Umami amino acid | 0.015 | 0.904 | 5.181 | 0.052 | 2.447 | 0.156 |
| Bitter amino acid | 2.006 | 0.194 | 51.460 | 0.000 | 0.725 | 0.419 |
| Aromatic amino acids | 1.037 | 0.338 | 34.252 | 0.000 | 0.894 | 0.372 |
| Sweet amino acid | 0.681 | 0.433 | 26.727 | 0.001 | 0.790 | 0.400 |
| Proportion of umami amino acids | 0.714 | 0.423 | 1.312 | 0.285 | 2.785 | 0.134 |
| Proportion of bitter amino acid | 0.083 | 0.780 | 1.714 | 0.227 | 0.383 | 0.553 |
| Proportion of aromatic amino acids | 0.028 | 0.870 | 0.087 | 0.775 | 0.519 | 0.492 |
| Proportion of sweet amino acid | 0.036 | 0.854 | 0.077 | 0.789 | 4.108 | 0.077 |

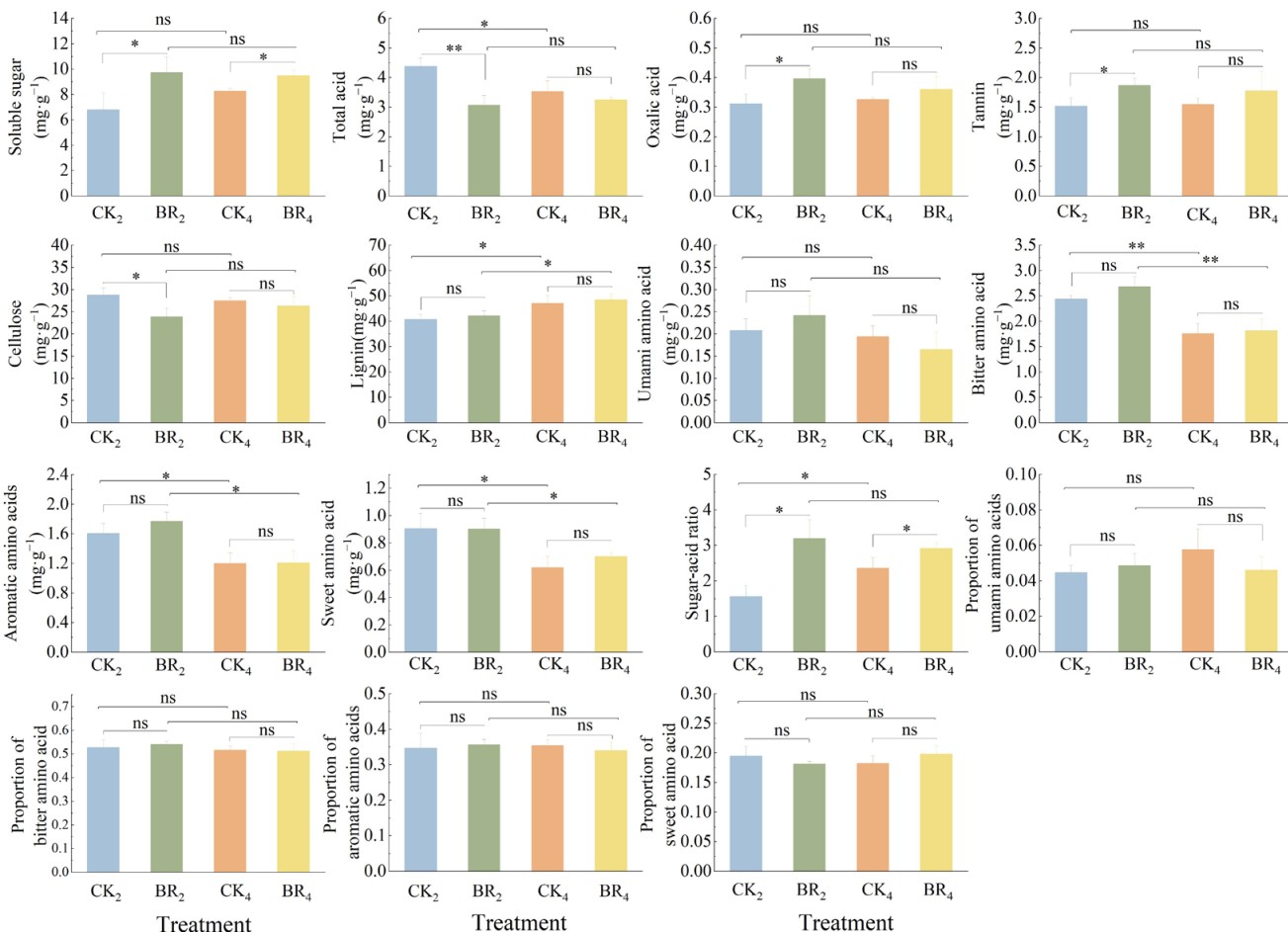

**Figure 5.** Flavor and roughness substances of *Ph. violascens* shoots after sheath blade removal. $CK_2$: natural growth after 2 d; $BR_2$: sheath blade removal after 2 d; $CK_4$: natural growth after 4 d; $BR_4$: sheath blade removal after 4 d; ns: $p > 0.05$; *: $p < 0.05$; **: $p < 0.01$.

## 4. Discussion

As *Ph. violascens* shoots emerged over time, the shoot length, individual weight, and base diameter all increased; however, the opposite effect was true for the edible rate of shoots, which was consistent with results on *Ph. prominens* obtained by Shi et al. [18]. These findings show that the apical meristem and internode tissue of *Ph. violascens* continue to grow and divide after shoot emergence, promoting high shoot rate growth [19,20]. Moreover, bamboo shoot cellulose and lignin content gradually increased as the growth period lengthened, while the lignification degree increased and the edible rate decreased [21]. The length, basal diameter, and individual shoot weight of the bamboo shoots decreased after sheath blade removal, and length significantly decreased 4 d after treatment. This could be because the non-structural carbohydrate (NSC) components of sheath blades are mainly composed of starch, which is the long-term energy storage source of plants. During rapid growth, starch can be converted into soluble sugar, which provides some energy for bamboo shoot growth. However, the resource acquisition and utilization capacity of bamboo shoots decreased after sheath blade removal, limiting the height growth process to a certain extent.

Nutritional components of bamboo shoots are key indices in the evaluation of quality. Results from this study showed that the content of protein, starch, total amino acid, and *EAA*s and the ratio of *EAA*s in bamboo shoots were not significantly affected by sheath blade removal. However, protein, total amino acid, and *EAA* content all decreased over treatment time. Protein is the main component of plant protoplasm. During the rapid bamboo shoot growth period, internode cells are constantly dividing and elongating, and

protein content is constantly consumed [22,23]. The protein consumption rate of bamboo shoots increased significantly after sheath blade removal, while protein content significantly decreased 4 d after treatment. The nutritional value of proteins is determined by the content, proportion, and availability of essential amino acids [24–26]. Studies that have evaluated the nutritional value of bamboo shoot proteins have found that the total essential amino acid score of proteins increased after sheath blade removal. It can therefore be concluded that sheath blade removal had no significant effect on the protein content in bamboo shoots while it had a certain effect on the *EAA*s of protein components. Egg proteins are approximately comparable to the human amino acid pattern [27,28]. *CD* characterizes the closeness between sample proteins and egg proteins. The closer that *CD* is to 1, the higher the nutritional value of proteins will be [29]. *EAAI* is an important indicator used to measure the amino acid balance of sample proteins. The closer that *EAAI* is to 100, the higher the protein quality and the utilization rate will be [21]. After sheath blade removal, protein *CD*, *EAAI*, and *NI* values in bamboo shoots increased, which indicated that the nutritional value of proteins improved to a certain extent by sheath blade removal, while proteins were more easily absorbed and utilized.

Bamboo shoot taste is determined by sugar, total acid, oxalic acid, tannic acid, fiber, and amino acid content [3,30], but the taste is likely to change as a result of the special compound or compounds which determine taste. In this study, we did not use organoleptic assay to quantify the parameter taste; rather, the taste was implied based on the constituents. Organoleptic assay is an important method for shoot assay of fruit and vegetables, but the method omits the importance of material base, particularly for shoot quality. Thus, this paper focuses on the shoot quality and the material base. We observed that sheath blade removal had no significant effect on the content and proportion of amino acid flavor compounds in bamboo shoots, but the soluble sugar content and the sugar–acid ratio increased significantly, indicating sheath blade removal was helpful in improving bamboo shoot sweetness. Soluble sugar provides the energy for the division and elongation of parenchyma within bamboo shoots [31], while sheath blade removal inhibits height growth processes and reduces the consumption of soluble sugar to some extent. Tannic acid is an economical and effective defense mechanism used by plants to guard against external injury [32,33]. Oxalic acid can regulate the osmotic potential while enhancing plant stress resistance [34]. Here, we found that bamboo shoots resisted damage caused by increasing oxalic and tannic acid content 2 d after sheath blade removal [33]. Additionally, we observed the existence of a dynamic balance between induced defense and growth investment, avoiding the overconsumption of resources under injurious or stress conditions [35,36]. Although sheath blade removal could induce an increase in tannic and oxalic acid content in bamboo shoots, it could not always maintain such high concentrations in resisting stress. The tannic and oxalic acid content in bamboo shoots returned to normal levels 4 d after sheath blade removal. Moreover, no significant comparative differences were observed with the control group 4 d after sheath blade removal. Cellulose is the main compound of plant cell walls. During the rapid bamboo shoot growth stage, cell walls quickly elongate and cellulose content quickly increase [37]. Sheath blade removal inhibited the height growth of bamboo shoots, resulting in a decrease in cellulose content and an increase in lignin content. This could closely correlate to an increase in lignin content, which thickens cell walls and increases the mechanical strength of bamboo shoots [38], thus enhancing its ability to resist stress.

## 5. Conclusions

In this study, sheath blade removal significantly inhibited the height-growth of *Ph. violascens* shoots and increased the nutritional value of proteins to a certain extent, while the nutritional quality and value of proteins significantly decreased as treatment time progressed. However, we detected significant differences in taste quality during different treatment times. For example, soluble sugar and acid astringent compounds (i.e., oxalic and tannic acid) significantly increased 2 d after sheath blade removal, but roughness obviously

decreased. Sweetness significantly increased 4 d after treatments, while the taste quality obviously improved. This showed that sheath blades are an important organ affecting bamboo shoot taste quality, which is mainly driven by carbon metabolism, but no obvious effect on nitrogen metabolism was observed.

**Author Contributions:** Conceptualization, S.C., Z.G. and S.X.; methodology, S.X., S.C., Z.G. and Y.H.; formal analysis, S.X., L.Y., Y.D., Y.X. and J.Z.; investigation, S.X., Y.H. and L.Y.; writing—original draft preparation, S.X. and S.C.; writing—review and editing, S.X., S.C. and Z.G.; funding acquisition, Z.G. All authors have read and agreed to the published version of the manuscript.

**Funding:** This research was funded by the Public Welfare Program of Zhejiang Province, grant number LGN22C160015; the Zhejiang Forestry Science and Technology Extension Project, grant number 2022B03 and National Natural Science Foundation of China, grant number 31770447.

**Institutional Review Board Statement:** Not applicable.

**Informed Consent Statement:** Not applicable.

**Data Availability Statement:** Data are contained within the article.

**Conflicts of Interest:** The authors declare no conflict of interest.

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
