# Peer review of "Effect of Sheath Blade Removal on Phyllostachys violascens Shoot Quality"

_agriculture, doi:10.3390/agriculture12091396_

Round 1

Reviewer 1 Report

The article is well written.

1. The parameter 'taste' was not quantified using an organoleptic assay. In this case, the 'taste' was implied based on the constituents. The authors should indicate that taste is likely to change as a result of the specific compound or compounds which determine taste.

2. Please include images documenting the progress of the experiment. The authors may refer to the following article as an example of graphical usage: https://www.mdpi.com/1999-4907/13/1/31

3. Abstract. The last sentence should state the implications for research and the future direction of the study.

4.  Line 89: The soil type must be described using an appropriate term (lateritic /loamy /clay). Red is a color and not an indication of soil type.

5. Line 91: Please indicate if a license was obtained from the relevant authorities for a sample collection from the forest. This is the standard procedure in most countries.

6. Line 100: (high) refers to tall.

7. LIne 109: What is meant by "edible weight"? Was any specific method applied to determine what is edible and what is non-edible?

8. Line 133: Does "closeness degree" refer to similarity? The term is not clear. Please rebut the reviewer's comment if this term is generally used in the analysis.

9. The term "treatment duration" is not clear on the first reading of the manuscript. The term treatment may mislead the reader to infer that there was an additional treatment such as a chemical treatment. Suggest the usage of the term "duration".

10. Reference (14): did the authors use an identical method in terms of instrumentation, column, detector, mobile phase, and instrument? This must be clarified because different laboratories may have different instruments. State the variable in the material and methods section: instrument, column, detector, flow rate, mobile phase, and any derivatization which may have been done.

Reviewer 2 Report

In the introduction, while raising the question:  74 Is there an obvious positive or negative change in P. violascens shoot quality during different 75 periods following sheath blade removal? Please mention which quality? Chemical or physical

Methodolgy: The treatment given is not very clear, please make it clear. Also please mention how were diffrent amino scids like umami, sweet, bitter etc measured. Give proper reference and make of the instruments used. 

In the results section, 3.2, 3.3 and 3.4 sections look overlaping. Seperate them either on the basis of physical and chemical parameters or make different sections for flavor, taste, chemical composition etc

Round 2

Reviewer 2 Report

It can now be accepted as most of the things have been addressed by the author